# Three-Dimensional-Printed Mechanical Transmission Element with a Fiber Bragg Grating Sensor Embedded in a Replaceable Measuring Head

**DOI:** 10.3390/s22093381

**Published:** 2022-04-28

**Authors:** Piotr Lesiak, Konrad Pogorzelec, Aleksandra Bochenek, Piotr Sobotka, Karolina Bednarska, Alicja Anuszkiewicz, Tomasz Osuch, Maksymilian Sienkiewicz, Piotr Marek, Michał Nawotka, Tomasz R. Woliński

**Affiliations:** 1Faculty of Physics, Warsaw University of Technology, Koszykowa 75, 00-665 Warsaw, Poland; konrad.pogorzelec.stud@pw.edu.pl (K.P.); aleksandra.bochenek.stud@pw.edu.pl (A.B.); piotr.sobotka@pw.edu.pl (P.S.); karolina.bednarska.dokt@pw.edu.pl (K.B.); tomasz.wolinski@pw.edu.pl (T.R.W.); 2Faculty of Electronics and Information Technology, Institute of Electronic Systems, Warsaw University of Technology, Nowowiejska 15/19, 00-665 Warsaw, Poland; alicja.anuszkiewicz@pw.edu.pl (A.A.); tomasz.osuch@pw.edu.pl (T.O.); 3Institute of Microelectronics and Photonics, Lukasiewicz Research Network, al. Lotników 32/46, 02-668 Warsaw, Poland; 4National Institute of Telecommunications, Szachowa 1, 04-894 Warsaw, Poland; 5Faculty of Power and Aeronautical Engineering, Warsaw University of Technology, 00-665 Warsaw, Poland; msienkiewicz@meil.pw.edu.pl (M.S.); pmarek@meil.pw.edu.pl (P.M.); 6Central Office of Measures, Elektoralna 2, 00-139 Warsaw, Poland; michal.nawotka@gum.gov.pl

**Keywords:** 3D-printed materials, optical fiber sensors, Fiber Bragg Gratings, compliant mechanisms

## Abstract

Compliant mechanisms have gained an increasing interest in recent years, especially in relation to the possibility of using 3D printers for their production. These mechanisms typically find applications in precise positioning systems of building robotic devices or in sensing where they can be used to characterize displacement. Three-dimensional printing with PLA materials allows fiber optic-based sensors to be incorporated into the structures of properly designed compliant mechanisms. Therefore, in this paper, an innovative technology is described, of a Fiber Bragg Grating (FBG) sensor embedded in a measuring head which was then inserted into a specially designed mechanical transmission element. The shape of this element is based on clippers that allow to freely modify the amplification of displacement amplitude so that the FBG sensor always works in the most optimal regime without any need to modify its external dimensions. Flexural sensitivity of the replaceable measuring head equal to 1.26 (mε/mm) can be adapted to the needs of the flexure design.

## 1. Introduction

In recent years, 3D printing has become one of the fastest growing industries, with applications ranging from construction industry to medicine [1,2,3,4,5,6,7,8,9,10,11,12,13,14,15,16,17,18,19,20,21]. Three-dimensional printing is the process of producing three-dimensional objects based on a computer model [11]. The essence of this process is building an element by overlapping thin layers of material which is also a binder [10,11]. Layers in printers are typically between 0.014 mm and 0.1 mm thick, which allows for extremely precise curves to be created [11].

The material used in 3D printers is in the form of a thin tape melted in the nozzle and spewed out as droplets with a diameter approximately corresponding to the thickness of the layer characteristic for a given device [14]. The typical temperature during 3D printing does not exceed 300–400 °C [18]; therefore, it is possible to embed a fiber-optic sensor in the element being made [12,13,14]. The layer made in this way around the fiber-optic sensor, on the one hand, will provide it with necessary protection allowing to place the fiber-optic sensor in any element and, on the other hand, it will allow for the measurement of stresses arising in the produced element. There are known cases of printing both optical fibers made of silica glass and polymers [14,15,16,17,18,19,20,21,22,23,24].

The combination of 3D printing with fiber-optic sensors opens up a wide range of new applications [15,16,17,18,19,20,21,22,23,24,25,26]. Compliant mechanisms obtain some or all of their movement due to the elasticity of individual members [6,7,8,9]. The compliant mechanism may comprise rigid body members and its movement can be achieved by connections formed between such components or can be produced as a single compliant component [9]. By placing the fiber-optic sensor at the most vulnerable points of the compliant mechanism, the deformation can be measured simultaneously with the movement while the component is constantly monitored [15]. Moreover, the fiber-optic sensor can be set in such a place that the deformation can be measured with an appropriate gain [19].

Modern compliant mechanisms usually use elements which are connected by means of flexible joints or elastic arms forming basic physical elements such as: a differential amplifier, a lever, a triangular mechanical amplifier [2,9], etc. In recent years, many bridge and lever type compliant mechanisms have been developed for measuring and amplifying the measurement of strain and pressure [19]. Connecting the compliant mechanism with the fiber-optic sensor is not a major problem [18]. Solutions presented in the literature assumed simple gluing of the fiber-optic sensor directly to the arms deformed under the influence of pressure or force. This way the measuring heads were made with measurement sensitivity enhanced even threefold [15].

Unfortunately, the main problem in published cases was the lack of protection of optical fiber, which could easily be damaged [15,19]. Additionally, a significant weakness of such a system is a presence of adhesive connections due to material differences on the border between glue and PLA. One of the possible solutions to overcome this limitation in composite materials is to place the optical fiber inside the material structure. The main reason for this is that the sensor becomes an integral part of the measured element (smart structure) and consequently there is no need to stick it outside that would result in possible defects in this connection. Such a solution enables also protecting the most sensitive parts of the sensor against any mechanical damages and at the same time allows deformation measurement. Therefore, in this work we focused only on designing a single compliant mechanism, which allows for complete printing of the fiber-optic sensor.

Most of the compliant mechanisms can be described as a single part capable of transmitting force and displacement due to its elasticity [5,6,7]. In order to design such a mechanism, it is necessary to use one of the two approaches to the problem [7]. The first approach, called kinematic-based, uses compliant elements that are represented as a rigid link [9]. In this approach, the force-to-displacement relations are determined using the Newton’s method. The second approach used in the present paper is a structural optimization-based approach. Hence, we have suggested a new topology, shape and size of the mechanism. Then, using numerical tools and by trial and error, the mechanism is optimized in terms of maximizing the measurement of deformation [1]. In addition, this paper will present a technology of safe printing of a fiber-optic sensor into a replaceable measuring head, which allows it to adjust the sensitivity of the sensor to deformation in order to optimize the measurement range. Inserting the optical fiber into the PLA material enables the optical fiber to be protected during deformation measurement.

## 2. Materials and Methods

### 2.1. Fiber Bragg Grating Based Sensor Fabrication

As a sensing element, fiber Bragg grating was chosen. Due to the fact that FBG relies on periodical modulation of the refractive index induced in the fiber core, it is an ideal candidate for integration with a structure fabricated by using a 3D printing technology. FBG was written in a standard, low-cost telecommunication optical fiber (SMF-28) ensuring compatibility with commonly used fiber-optic components and measuring equipment. However, due to inherently low Ge concentration in the fiber core, and thus, low efficiency of grating inscription in SMF-28 by UV illumination, the fiber was photosensitized prior to FBG recording [27]. To increase grating photosensitivity, the optical fiber was hydrogen loaded under 11.5 MPa of pressure at room temperature for a couple of days. After that, the fiber Bragg grating was successfully written by a krypton fluoride (KrF) pulsed excimer laser operating at 248 nm and phase mask technique that ensures the best possible repeatability and accuracy. Then the fiber Bragg grating with a resonance wavelength of c.a. 1538 nm was obtained, which matches the spectral range of a commonly used fiber component and instruments used for FBG interrogation. Finally, the fiber Bragg grating was thermally annealed at elevated temperature to ensure long term stability and resistance to short-term effects of higher temperatures during the 3D printing process.

### 2.2. Three-Dimensional Printing Technology

Firstly, the main focus was set on safe and stable placement of the fiber Bragg grating sensor in a 3D printed element. Therefore, a cuboid was chosen as the basic model of the element in which the optical fiber is embedded (Figure 1a). Then, the printing process was divided into two phases in such a way as to enable safe and stable placement of the optical fiber without damaging it. In the first phase, the lower part of the cuboid of 10 mm × 60 mm × 6 mm (width, length, height) was printed as a base on which the optical fiber with an in-written Bragg grating was placed. Then, the upper part of the cuboid was printed on the top of the optical fiber. This part should consist of at least two print layers to protect the optical fiber (Table 1). The printed element in the form of a test bar was prepared for further tests.

During a preliminary experimental study, it was found that an acrylate coating should be removed from the optical fiber before the second phase of printing. It ensures better strain transfer to the optical fiber with in-written FBG. Additionally, for a better and more reproducible FBG sensor arrangement the fabrication of v-groove in the base is needed. The depth and width of the v-groove, matched to the optical fiber diameter, is determined by the resolution of the 3D printer, which was 180 μm. The thickness of one print layer was 0.1 mm; therefore, the v-groove consisted of two layers.

The top of the test bar is shorter than the base. It allows for optimal fixing of the optical fiber to the edges of the base, ensuring its high mechanical strength. Namely, the central section of the optical fiber with an in-written FBG and removed acrylate coating is protected, while only the coated part of the optical fiber is fixed to the base and exposed to external factors.

The list of all elements related to 3D printing of a fiber-optic sensor with an FBG sensor is presented in Table 1.

Finally, it was decided that for the initial tests’ dimensions of the lower part of the test bar should be consistent with the following dimensions (width, length, height): 10 × 60 × 6 mm, and the upper part of the test bar should be consistent with the dimensions of 10 × 40 × 2 mm (Figure 1b). The replaceable measuring head (designed in a similar way to the test bar) should be redesigned and adapted to the final dimensions of the compliant mechanism. In order to reduce its stiffness, it was decided to decrease the number of printed layers. The total thickness of the replaceable measuring head was 1.5 mm (13 and 2 print layers on the lower and upper parts respectively) and the ultimate technology used to print the replaceable measuring head intended for the compliant mechanism was consistent with the technology used to print the first test bar.

### 2.3. Numerical Calculations

The finite element method (FEM) is a numerical method employed to solve the partial differential equation present in mathematical physics. An important step of modeling is the discretization of the analyzed domain, that is dividing the area into subdomains called finite elements. The solution is obtained in each and every one of them as a result of applying the proper boundary conditions. For the analysis presented further the ANSYS v19.2 software has been employed (Ansys® Academic Research Mechanical, Release 19.2.).

Considering the possibility of fiber-optic sensor placement within the printed measuring head in the shape of a rectangular bar, it was decided to prepare a completely new design of a uniform compliant mechanism, which was able to transfer the deformation on the principle of modified scissors inscribed in the shape of an ellipse (Figure 2a). In our type of the compliant mechanism, displacement (Δ*L*) is transferred to a three-point bending system, in which the rear wall of the mechanism acts as pressing element, and the replaceable measuring head with a fiber-optic sensor leans on rigid arms.

The prepared model was processed numerically. The aim of simulations was to check the principle of the compliant mechanism and then optimize it in terms of deformation transfer. For this analysis, a FEM model of the susceptible mechanism (Figure 2b) was prepared. Due to the double symmetry of the tested element, a quarter of the whole slice was considered. A simplification was introduced, consisting in the removal of an optical fiber channel—due to its size, the impact on the stiffness of the entire structure is negligible.

The initial finite element mesh is finer in the area of the element’s contact with the replaceable measuring head (Figure 2c), while it is coarser in the remaining areas. In addition to the introduced symmetry conditions on the appropriate surfaces, the element is supported at the point of contact with the other half, thus enabling the rotation of the entire arm. The displacement Δ*L* applied to the lower part of the arm was a consequence of the applied load. To model the interactions between the element and the replaceable measuring head, frictionless contact elements have been introduced in the contact zones.

Table 2 shows the material constants used in numerical calculations. Due to the possibility of easy printing of the optical fiber in PLA material, numerical calculations were limited to this material only.

The model of the replaceable measuring head was numerically tested separately—in the three-point bending system (Figure 3). Supports and the loading element are marked in blue. Their material set to non-deformable to model much stiffer experimental equipment. As a part of the numerical analysis, only the replaceable measuring head was tested, intended to be placed in the compliant mechanism with the dimensions ×1.5 mm.

### 2.4. Measurement Setup

A superluminescent diode (Thorlabs, Inc., Fiber Coupled SLD Source, Newton, NJ, USA), with a central wavelength of 1550 nm and spectral width of 100 nm, and an optical spectrum analyzer (Yokogawa Electric Corporation, AQ6370C, Tokyo, Japan) were used for the tests of the thick rectangular bar with an embedded FBG sensor. The light from the source was introduced into the optical fiber and fed to the sample through an optical circulator. The Bragg wavelength reflected from the FBG was re-routed to the circulator and then through its third port delivered to the OSA [28]. First, the rectangular bar was checked in the three-point bending system (Figure 4a), and then in the system with a Peltier device (Figure 4b). In the final measurement system (Figure 4c), the compliant mechanism was deformed by applying an appropriate force to the arms.

The compliant mechanism was printed by using a Prusa Research a.s., i3 MK3 printer, which was placed in a built-in chamber to avoid sudden differences at ambient temperatures. In this case, the thickness of the first layer 0.2 mm and the rest of the layers was 0.1 mm. The nozzle printer temperature was 225 °C, the base of the printer was 65 °C. Completion of the element is set to 100%.

## 3. Results

### 3.1. Numerical Calculation Results

Several models of the compliant mechanism configurations were analyzed numerically. They differ in the length of the replaceable measuring head (*b*), sample length (*d*), length of the element pressing against the sample (*k*) and the length of the lower edge (*r*), as presented in Figure 5. The load is the displacement in the horizontal direction of the lower element—Δ*L*. The same Δ*L* value, 3.5 mm, was selected for the model comparison.

Numerical calculations showed that the geometry of the compliant mechanism significantly influences the deflection of the replaceable measuring head. As a part of all numerical tests, the value of deflection *s* of the replaceable measuring head associated with the place of located optical fiber was calculated. This value is calculated at each step as the difference between the vertical displacement of the replaceable measuring head and the vertical displacement of the opening edge supporting the bending replaceable measuring head. The graphs of the deflection as a function of the 1/2 Δ*L* loading displacement for the change of individual parameters are presented in Figure 6a–d. Numerical calculations showed that the change in the length of the lower edge *r* has the greatest influence on the value of the deflection s. The calculations show that *s* strongly decreases with an increasing value of *r* (Figure 6a). Ultimately, the *r* value was 49 mm for the final element. The minimal value analyzed numerically strongly stiffened the structure; therefore, it was decided to adopt a higher value of *r* for further work. The procedure for selecting the values of the remaining quantities was different. Modifying only one element causes a change in the parameters of the ellipse, and therefore of the remaining elements. However, the performed numerical calculations demonstrated that each of the remaining elements has less and less influence on the value of *s*. The length of the element pressing the replaceable measuring head *k* was analyzed in the range from 20 mm to 50 mm (Figure 6b and a decrease in the value of the deflection *s* as a function of the length *k* was observed. Consequently, we chose a *k* value of 27 mm for the final element. The sample length was analyzed from 200 mm to 230 mm (Figure 6c). It was noticed that the deflection value *s* decreases as a function of *d*. We selected a *d* value of 200 mm for the final element. The last analyzed parameter was the length of the replaceable measuring head, with the analyzed range from 70 mm to 86 mm (Figure 6d). Unlike the previous cases, the value of the deflection *s* increases as a function of *b*. We chose a *b* value of 61 mm for the final element due to the fact that the stress acting on the optical fiber in the replaceable measuring head depends on the distance between the lower supports in the three-point bending system. If this distance increases, the stress decreases (Equation (1)).

Knowing the deflection values of the selected compliant mechanism, strain analysis in the replaceable measuring head was initiated (Figure 7a). It was assumed for the calculations that the central element will move vertically down by 1.10 mm. The values of normal stresses and normal strains were obtained from the area where the optical fiber would run. Due to the fact that the optical fiber is located in the compressed layer of the replaceable measuring head, the strains are negative and depend linearly on the deflection value of the sample (Figure 7b).

### 3.2. Measurement Results

#### 3.2.1. 3-Point Bending Measurement

A test bar having a total thickness of 8 mm was examined in the three-point bending measurement system. The FBG sensor was located in a compression layer at a distance of 2 mm from the inert layer. In the process of bending, it was observed that the Bragg wavelength decreased with deflection of the rectangular bar (Figure 8a). Since the FBG spectrum has a distorted top and it is difficult to track the position of the maximum value, the 3-dB bandwidth method to determine the Bragg wavelength was used [29]. In Figure 8b the dependence of the strain (modulus) on deflection is shown. In this figure, the experimental strain values (blue dots) were compared with the strain values calculated from the equation derived for the three-point bending system (red line) [30]:*ε* = 6*sw*/*h*^2^(1)
where *w* is the distance between fiber and middle of the sample and *h* is the distance between lower supports.

The comparison shows that the strain values calculated from Formula (1) are slightly larger than the experimental values. It follows that the FBG sensor measures the strain occurring in a given place of the rectangular bar. It can be concluded that during the 3D printing, no additional stresses appeared on the FBG sensor that would change its sensitivity [31]. This is the proof of a perfect connection between the FBG sensor and the boom. The sensitivity of the sensor can be determined from Figure 8b. and is equal to 9.36 mε/mm. On the other hand, this slight difference between the theoretical and experimental values may indicate plastic deformations appearing in the sample—the sample thickness was 8 mm. It should be noted that the differences increase with an increasing deflection. Therefore, in further works, a much thinner rectangular bar has to be designed to improve the accuracy of the deformation measurement.

#### 3.2.2. Temperature Measurement

To measure the effect of temperature on the printed FBG sensor in PLA material, the printed test sample was heated with a Peltier device. While heating the rectangular bar, it was observed that the Bragg wavelength increases with the ambient temperature of the rectangular bar. In Figure 9a, selected reflection characteristics of the printed FBG sensor are shown.

Typical sensitivity of the FBG sensor written in standard SMF-28 fiber to temperature (before printing) is c.a. 10 pm/K [32]. However, after the measurements were carried out, a high increase in temperature sensitivity was observed (Figure 9b) up to 87 pm/K. The reason for this may be the influence of the PLA material on the optical fiber [33]. The relatively high value of the thermal expansion parameter for PLA, equal to 41 μm/(m·K) [34], means that the optical fiber can be subjected to high stress despite the fact that it has been printed in a v-groove. The higher value of the sensitivity to temperature may result from the fact that the PLA material, when expanding, strongly presses on the optical fiber. Additionally, these stresses are heterogeneous in nature, which can be observed at higher temperatures (Figure 9a). As the temperature increases at the top of the curve, deformations and deviations from the predicted shape are visible, which is maintained for lower temperatures.

### 3.3. Compliant Mechanism Measurements

In the arrangement shown in Figure 4c, a compliant mechanism was tested with a replaceable measuring head having a total thickness of 1.5 mm. The FBG was covered with melted printing material (2 layers), as reported in Table 1. Theoretical sensitivity of such a sensor is much smaller than the sensor in the thick rectangular bar and equals 1.26 mε/mm. The FBG sensor was located in a compressed layer at a distance of 0.55 mm from the inert layer. During the application of deformation to the arms of the compliant mechanism, it was observed that the Bragg wavelength decreased with the strain mechanism applied to the compliant arms (Figure 10a). For simplicity, in Figure 10b dependence of the strain modulus on deflection is shown (blue dots).

The range of deformations of the compliant mechanism tested was within the range of elastic deformations. After releasing the Δ*L* deformations, the Bragg wavelength returned to its original value (Figure 11a). During the long-term stability measurement (1 h), the maximum temperature changes did not exceed ±0.5 °C, which translates into ±50 pm (taking into account the temperature sensitivity of the FBG embedded in PLA). Therefore, in Figure 11b, slight fluctuations for the individual measurement cycles are visible.

A comparison of the measured strains with a given displacement equal to 1 mm (presented in Figure 10b) with the numerically calculated deformation (presented in Figure 7b) is presented in Table 3. The higher value of the measured deformation compared to the numerical calculations may be influenced by the complex state of stresses arising in the replaceable measuring head at the point of contact with the pressing element [31,32]. There are only two PLA layers between this element and the FBG sensor; therefore, the FBG sensor may be subjected to additional stresses. In the case of the test rectangular bar, when there were 20 PLA layers between the element pressing the rectangular bar and the FBG sensor, this effect was not observed.

## 4. Discussion

The experimental results obtained show that the Bragg wavelength changed with the increasing applied strain for both: the test beam and the replaceable measuring head. The nature of these changes is linear confirming a good connection between the fiber-optic sensor and PLA material. The qualitative agreement of the experimental strain sensitivity with the numerically calculated value indicates that the actual compliant member is performing as predicted. Differences in the indentations may result from the complex state of stresses developed around the FBG sensor when applying pressure to the flexible element beam in a three-point bending system. In this sensor configuration, the replaceable measuring head is under stress in the immediate surroundings of the FBG sensors. However, depending on the needs, the fiber-optic sensor can be placed in both the compressed and stretched layer of the interchangeable measuring head. Placing the FBG sensor in the tension layer will increase the distance between the FBG sensor and the pressing element and will protect the sensor from a complex state of stress.

Additionally, temperature sensitivity of the printed FBG sensor was measured. The observed high thermal expansion of the PLA material (Figure 9b) causes a significant increase in temperature sensitivity of the FBG sensor placed in the rectangular bar, especially since the sensor was placed in the PLA material without any coating. To reduce this impact, it is necessary to consider such coverage of optical fiber to minimize the impact of the material on the optical fiber. However, such a coating must be resistant to the temperature of the 3D printer head so that it is not damaged during the printing process.

The sensitivity of the replaceable measuring head can be easily modified by changing its thickness (changing the number of layers in the printout). The flexural sensitivity of the rectangular bar was 9.36 (mε/mm) and the replaceable measuring head 1.26 (mε/mm). Thus, there is a wide range of possibilities to adapt the sensitivity of the fiber-optic sensor to the needs of the flexure design.

## 5. Conclusions

A new design of the compliant mechanism presented in this paper allows easy modification of the parameters such as thickness and length of the replaceable measuring head, sample length, length of the element pressing against the sample, and the length of the lower edge. Such degrees of freedom allow easy adjustment of measuring range to maximum sensitivity of the FBG sensor. As a result of the design work carried out, it has been shown that the most optimal shape of the flexible element with optimal parameters will be an ellipse. Additionally, the paper presents the technology of safe FBG sensor printing into PLA material during the process of 3D printing. The fiber-optic sensor has been printed in the shape of a rectangular bar, which was easily adapted to the requirements of a replaceable measuring head intended to be attached to the mechanical transmission element.

## Figures and Tables

**Figure 1 sensors-22-03381-f001:**
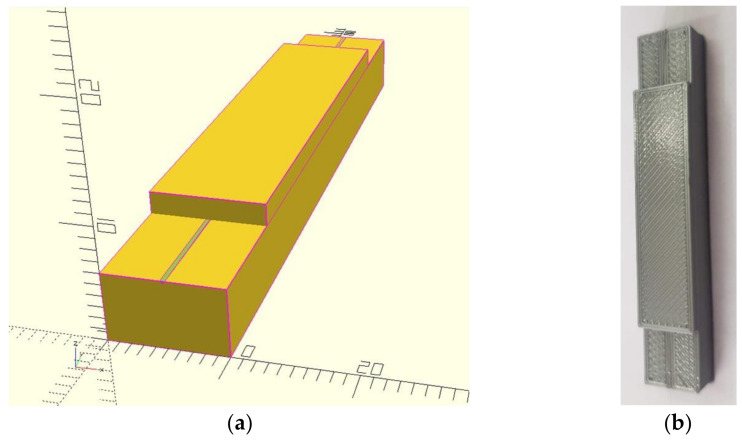
Three-dimensional model of the test bar with a v-groove intended for the optical fiber in a mm scale (**a**) and the bar printed for initial tests (**b**).

**Figure 2 sensors-22-03381-f002:**
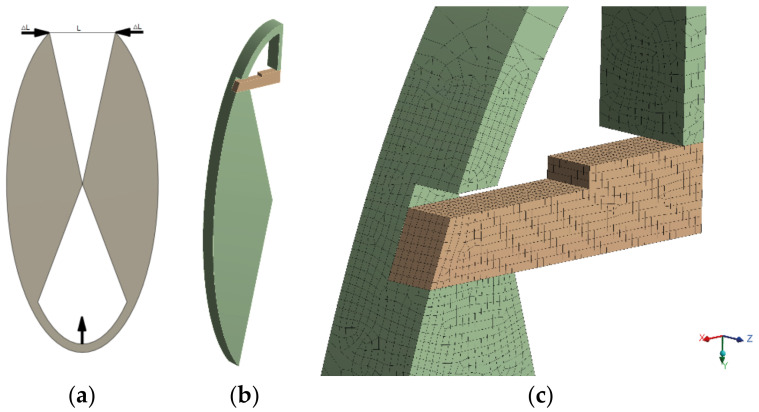
Model of a compliant mechanism (**a**), the geometry of the modeled object. Orange—rectangular bar, green—element (**b**) and view of the mesh refinement in the area of element and rectangular bar contact (**c**).

**Figure 3 sensors-22-03381-f003:**
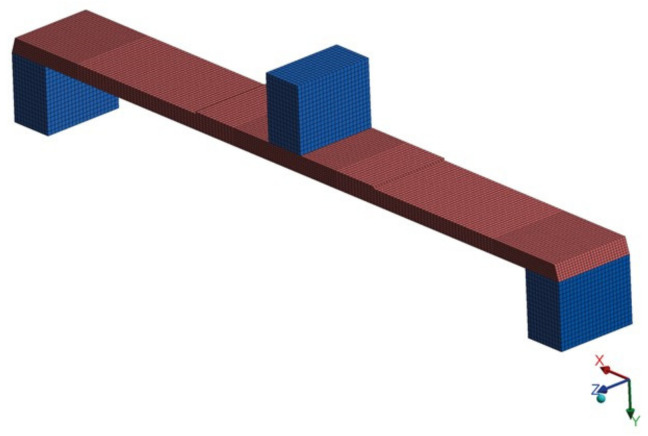
Numerical model of a replaceable measuring head intended for the compliant mechanism.

**Figure 4 sensors-22-03381-f004:**
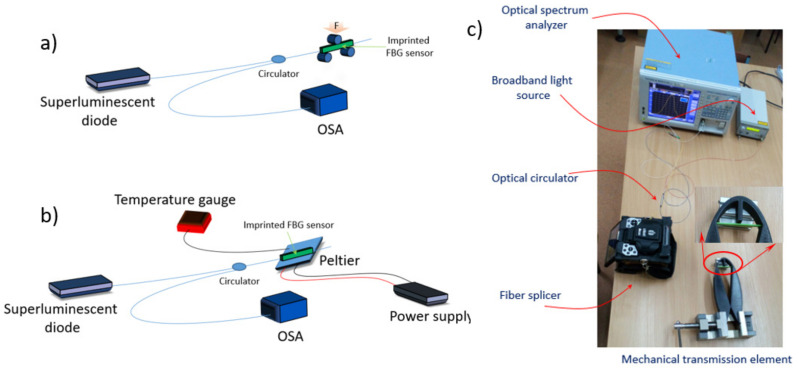
Measurement setups for strain (**a**), temperature (**b**) and compliant mechanism (**c**).

**Figure 5 sensors-22-03381-f005:**
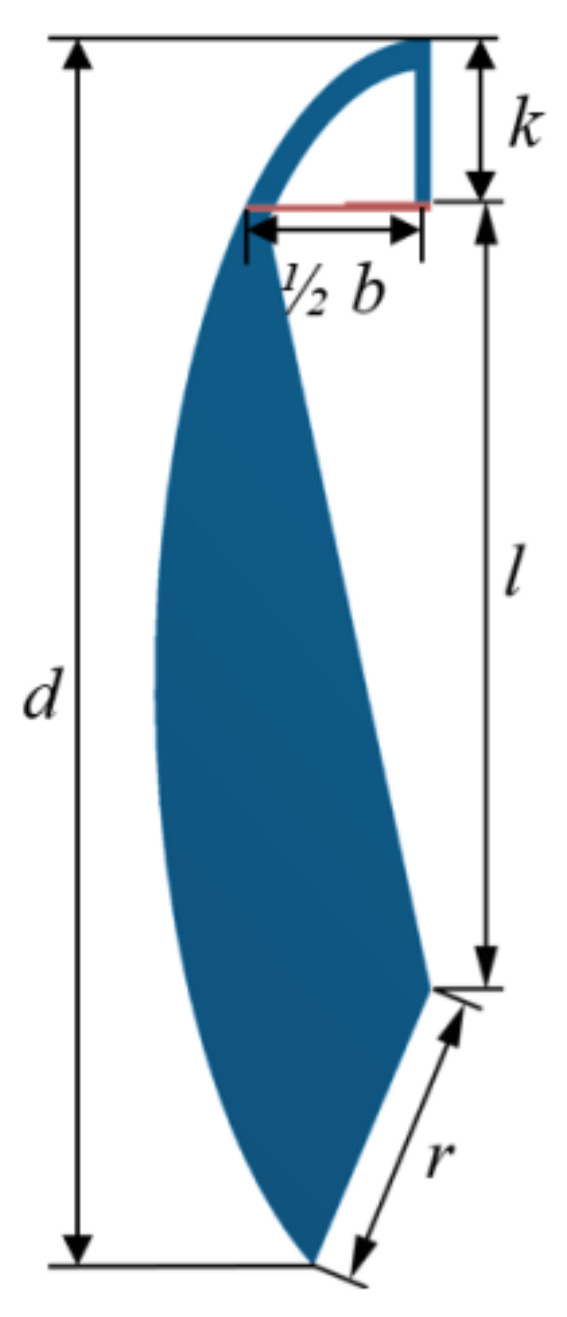
Models analyzed numerically: the scheme parameters.

**Figure 6 sensors-22-03381-f006:**
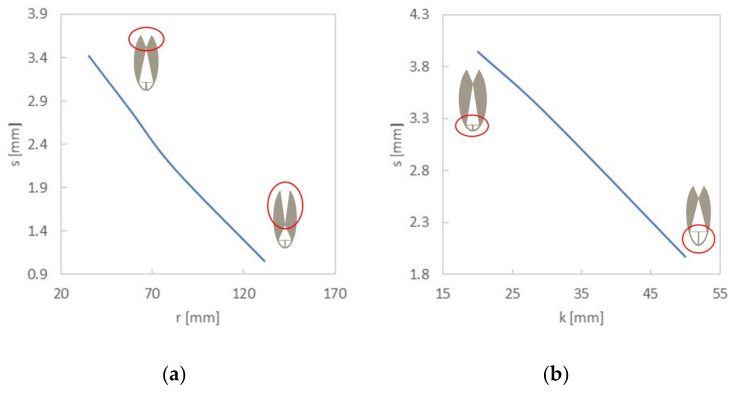
Numerical calculations results. The individual graphs present the dependence of the deflection of the replaceable measuring head on the length of the lower edge (**a**), length of the element pressing against the sample (**b**), sample length (**c**) and length of the replaceable measuring head (**d**).

**Figure 7 sensors-22-03381-f007:**
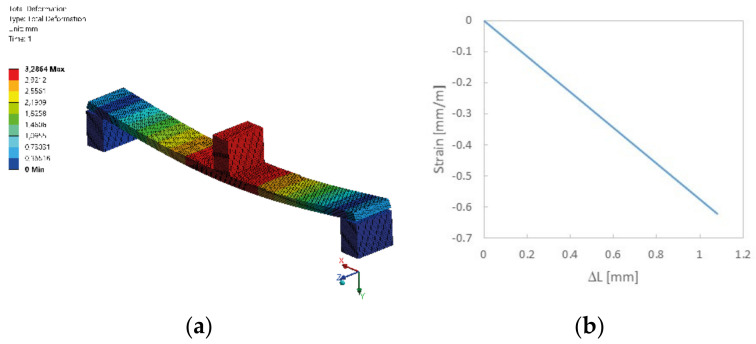
Replaceable measuring head model in a three-point bending system (**a**) and the results of the numerical strain simulation at the location of the fiber (**b**).

**Figure 8 sensors-22-03381-f008:**
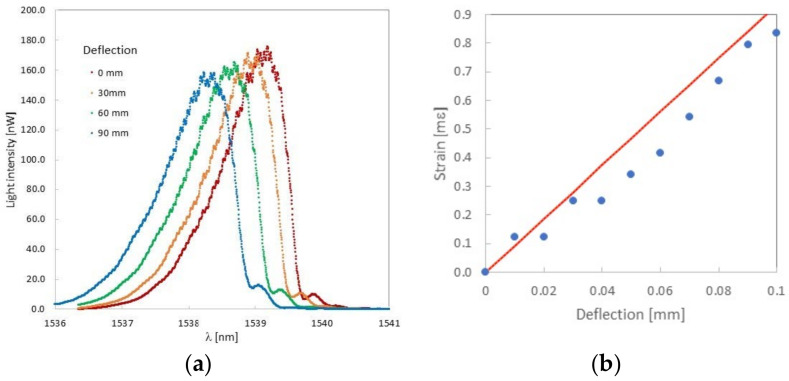
Change of the Bragg wavelength as a function of deflection (**a**) and comparison of the measured strain (modulus) value with the theoretical value (**b**) for a rectangular bar with a thickness of 6 mm.

**Figure 9 sensors-22-03381-f009:**
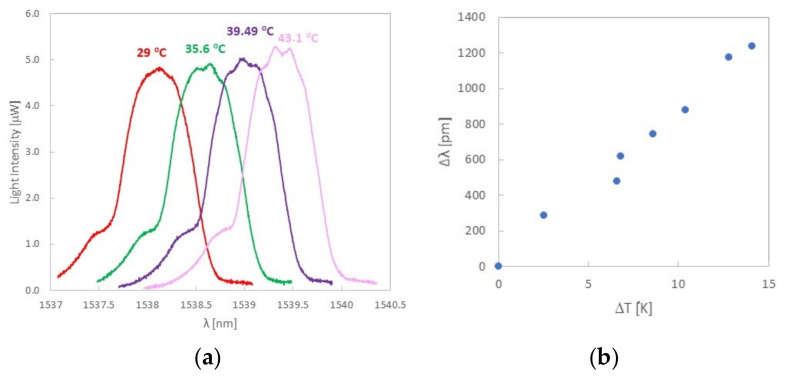
Change of the Bragg wavelength as a function of temperature (**a**) and determination of temperature sensitivity (**b**) for a rectangular bar with a thickness of 6 mm.

**Figure 10 sensors-22-03381-f010:**
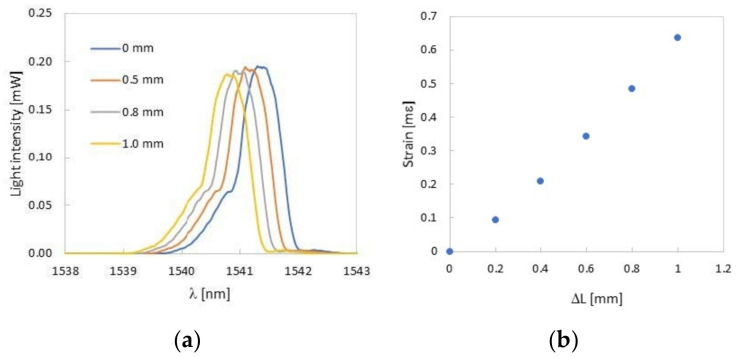
Change of the Bragg wavelength as a function of Δ*L* (**a**) and determination of strain (modulus) sensitivity (**b**) for a replaceable measuring head with a thickness of 1.5 mm placed in the compliant mechanism.

**Figure 11 sensors-22-03381-f011:**
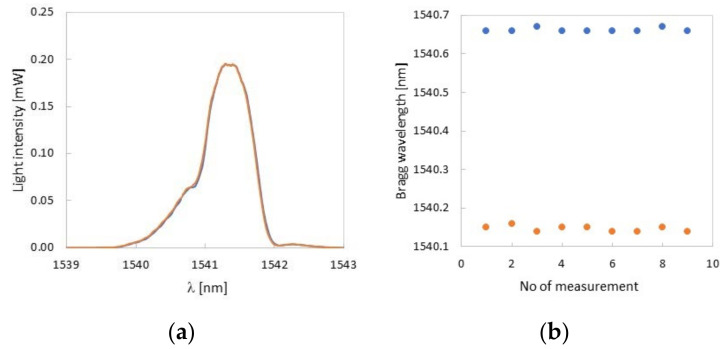
Measurement repeatability test (**a**) and time stability test (**b**).

**Table 1 sensors-22-03381-t001:** Three-dimensional printing main parameters.

**Optical Fiber**
With partially removed coating
**3D Printer**
Programmed change of filament color as a pause
Resolution limitation—180 µm v-groove
**Printing Process**
Supports for optical fibers Stretching the optical fiber during printing
Nozzle printer temperature: 215–220 °C; table: 60 °C
At least two layers covering the optical fiber
Fan power reduction to 60% after the 2nd layer

**Table 2 sensors-22-03381-t002:** Materials parameter used in numerical calculations.

	PLA
Young modulus E (MPa)	3170
Poisson coefficient	0.331

**Table 3 sensors-22-03381-t003:** Comparison between measured and calculated strain sensitivity of the replaceable measuring head.

	Strain Sensitivity (mε/mm)
Measured	0.64
Calculated	0.59

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
