# Peer review of "Three-Dimensional-Printed Mechanical Transmission Element with a Fiber Bragg Grating Sensor Embedded in a Replaceable Measuring Head"

_sensors, 2022, doi:10.3390/s22093381_

Round 1

Reviewer 1 Report

In the manuscript '3D-printed mechanical transmission element with a Fiber Bragg Grating sensor embedded in a replaceable measuring head', the authors used 3D printing method to improve the fiber's mechanical properties such that it can afford more deformation. However, the authors did not provide the experimental results showing the improvement of the mechanical strength before and after 3D printing. Thus it would be unconvincing to the readers to believe the impact and necessity of 3D printing. This reviewer would also suggest the authors provide a cross-section image of the fiber before and after. Therefore, this reviewer believe that this manuscript can be considered after major revision.

Author Response

Thank you very much for valuable and insightful comments. All comments were taken into consideration and the text of the manuscript has been corrected.

In the manuscript '3D-printed mechanical transmission element with a Fiber Bragg Grating sensor embedded in a replaceable measuring head', the authors used 3D printing method to improve the fiber's mechanical properties such that it can afford more deformation. However, the authors did not provide the experimental results showing the improvement of the mechanical strength before and after 3D printing. Thus it would be unconvincing to the readers to believe the impact and necessity of 3D printing. This reviewer would also suggest the authors provide a cross-section image of the fiber before and after. Therefore, this reviewer believe that this manuscript can be considered after major revision.

Thanks a lot for your review. The aim of our work was to develop a new compliant mechanism with a replaceable measuring head. This head has a built-in optical fiber that allows to measure the deformation appearing in the head. Inserting the optical fiber into the PLA material allows to protect the optical fiber during the measurement of deformation. However, the mechanical strength of the optical fiber has not changed. Our goal was not to improve the mechanical strength of the optical fiber before and after 3D printing, therefore there are no research results related to this.. Additionally, we do not expect any changes to the cross-section image of the fiber before and after 3D printing. Temperature in the 3D printer is too low (225 °C) to affect the shape of the silica glass fiber. Therefore, we did not conduct research in this area.

Reviewer 2 Report

This paper presents the results of a study of a part made by 3D printing with an integrated fiber Bragg grating. A transducer was made to study the mechanical properties of the part.
The authors have done interesting work, however, in my opinion, it has little novelty and poor elaboration.
I will note a number of shortcomings.
- the manufactured part (Figure 1) does not correspond to the one that was included in the three-point model to assess its deformation (Figure 3);
- the FBG spectrum (Figure 8) has a distorted top, the algorithm for calculating the central wavelength of the grating built into the spectrum analyzer you use defines it as a simple maximum, which for such a distorted top will give a significant error that has not been estimated;
- it is not clear what caused the choice of a linear approximation for the measurement results (Figures 8b, 9b, 10b), an estimate of the approximation accuracy is not given, in addition, the amount of data is too small;
- it is known that the FBG response depends on both temperature and deformation; therefore, it is not clear how this dependence was taken into account when measuring deformation (Figures 8b and 10b) and long-term stability (Figure 11);
- the repeatability of the response after several load cycles has not been studied;
- the proposed method of introducing an optical fiber into a part is not suitable for practical use in strain control, because there are no mechanisms for compensating for the temperature drift of the grating

Author Response

Thank you very much for valuable and insightful comments. All comments were taken into consideration and the text of the manuscript has been corrected.

This paper presents the results of a study of a part made by 3D printing with an integrated fiber Bragg grating. A transducer was made to study the mechanical properties of the part.
The authors have done interesting work, however, in my opinion, it has little novelty and poor elaboration.
I will note a number of shortcomings.
- the manufactured part (Figure 1) does not correspond to the one that was included in the three-point model to assess its deformation (Figure 3);

Answer: In Fig. 1, the test sample (test bar) is shown. For the calculations, we assigned an optimized sample (replaceable measuring head), that is why they differ from each other. Table 3 compares the results of numerical and experimental calculations only for the replaceable measuring head.

- the FBG spectrum (Figure 8) has a distorted top, the algorithm for calculating the central wavelength of the grating built into the spectrum analyzer you use defines it as a simple maximum, which for such a distorted top will give a significant error that has not been estimated;

Answer: In all cases, we used the 3-dB bandwidth method to determine the Bragg wavelength. The description in the manuscript has been completed and an appropriate reference has been added:

Since the FBG spectrum has a distorted top and it is difficult to track the position of the maximum value, the 3-dB bandwidth method to determine the Bragg wavelength was used [30].

Tosi, D. Review and Analysis of Peak Tracking Techniques for Fiber Bragg Grating Sensors. Sensors 2017, 17, 2368. https://doi.org/10.3390/s17102368

- it is not clear what caused the choice of a linear approximation for the measurement results (Figures 8b, 9b, 10b), an estimate of the approximation accuracy is not given, in addition, the amount of data is too small;

Answer:  Fig. 8b does not contain a linear approximation fitted to the measurement points. In this figure we have a comparison of the experiment with the theoretical value of stresses calculated from formula 1. It has been additionally described in the text:

In Fig. 8b the dependence of the stress modulus on deflection is shown. In this figure, the experimental stress values (blue dots) were compared with the stress values calculated from the equation derived for the 3-point bending system (red line) [31]:

Answer: We agree with the reviewer that the amount of data in Figures 9b and 10b is too small. Linear approximation has been removed. The temperature and strain sensitivity for further analysis was taken as the average value.

- it is known that the FBG response depends on both temperature and deformation; therefore, it is not clear how this dependence was taken into account when measuring deformation (Figures 8b and 10b) and long-term stability (Figure 11);

Answer:  During the measurements shown in Figs 8b and 10b temperature was constant at 20 +/- 1 oC. Also, during long term stability (1h) the maximum temperature changes did not exceed +/- 0.5K, which translates into +/- 44 pm (taking into account temperature sensitivity of the FBG embedded in PLA).

- the repeatability of the response after several load cycles has not been studied;

Answer:  The manuscript was supplemented with additional studies (Fig. 11b).

- the proposed method of introducing an optical fiber into a part is not suitable for practical use in strain control, because there are no mechanisms for compensating for the temperature drift of the grating

Answer:  The idea of the paper is to present an interesting solution as a proof of concept, and not its research necessary to create a technical specification. Hence the lack of full testing studies (not this stage yet). The issue of temperature compensation can be solved simply with a second grid (known from the literature). 

Reviewer 3 Report

The paper reports the possibility and the effects of a FBG embedded in a measuring head.

The work is interesting, but some improvements need to be done and some issues need to be addressed.

Abstract

  • Some details about the results should be provided.

Introduction

  • Nevertheless the state-of-the-art is robust, the synthetic description of the aim of the work is insufficient.

Materials and methods

  • Line 92 – Replace bars with bar.
  • It should be emphasized that the FBG was covered with melted printing material (2 layers), as reported in tab 1.
  • Line 116 – Replace micrometers with µm.
  • Fig 1 – The units of the axis should be reported.
  • Line 136 - If you change the number of layers, the thickness of the single layer will also change. This seems inconsistent compared to what was previously stated (line 109: This part should consist of at least two print layers to protect the optical fiber.). Please, provide an explanation.
  • Line 152 and 165 – The same symbol ΔL was used for deformation and displacement. Please, provide an explanation.
  • It is necessary to provide all the details relating to the setup (brand and model or characteristics of each device), so that the reader can replicate the described experiments.
  • Line 201 – Replace degrees C with °C.

Results

  • mstrain is not a unit of the International System of Units.
  • Fig 8 b) – Indicate a description for the red line.

Author Response

Thank you very much for valuable and insightful comments. All comments were taken into consideration and the text of the manuscript has been corrected.

The paper reports the possibility and the effects of a FBG embedded in a measuring head. The work is interesting, but some improvements need to be done and some issues need to be addressed.

Abstract
· Some details about the results should be provided.
Answer: The description has been completed

Introduction
· Nevertheless the state-of-the-art is robust, the synthetic description of the aim of the work is insufficient.

Answer: The description has been completed

Materials and methods
· Line 92 – Replace bars with bar.

Answer: The word has been changed

· It should be emphasized that the FBG was covered with melted printing material (2 layers), as reported in tab 1.

Answer: The sentence was added in line 316.

· Line 116 – Replace micrometers with μm.

Answer: The word has been changed

· Fig 1 – The units of the axis should be reported.

Answer: The description of the Fig 1 has been changed

· Line 136 - If you change the number of layers, the thickness of the single layer will also change. This seems inconsistent compared to what was previously stated (line 109: This part should consist of at least two print layers to protect the optical fiber.). Please, provide an explanation.

Answer: Changing the number of printed layers does not change the thickness of a single layer. The thickness of a single layer is always 0.1 mm.

· Line 152 and 165 – The same symbol ΔL was used for deformation and displacement.
Please, provide an explanation.

Answer: symbol ΔL has been assigned to displacement only.

· It is necessary to provide all the details relating to the setup (brand and model or characteristics of each device), so that the reader can replicate the described experiments.

Answer: Description has been added

Line 201 – Replace degrees C with °C.
Answer: Description has been added

Results
· mstrain is not a unit of the International System of Units.

Answer: Symbol mstrain has been changed.

· Fig 8 b) – Indicate a description for the red line.

Answer: Description has been added

Reviewer 4 Report

This paper reports a 3D-printed mechanical transmission element with a FBG sensor embedded in a replaceable measuring head. I have some comments.

1- Introduction: please consider other literature that is missing about 3D using optical fiber sensores not only in silica fiber but also in polymer. Materials 11 (11), 2305, 2018; Sensors 19 (16), 3514, 2020. among others.

2. Fig. 8 to 10: please consider to add more cycles, increasing and decreasing cycles to confirm the repeatability. Also, Fig. 8a and Fig. 9a the FBG are the same? Why the spectra are so different? 

3- Fig. 11: it is not a good graph to expain the repeatability. It is just confirm the that spectrum comes to the initial position. Please comment.

4- Table 3. What is "nº 15"?  Also the principle for such different is not so clear.

5. How many probes were tested to guarantee the reproducibility between identical probes?

Author Response

Thank you very much for valuable and insightful comments. All comments were taken into consideration and the text of the manuscript has been corrected.

This paper reports a 3D-printed mechanical transmission element with a FBG sensor embedded in a replaceable measuring head. I have some comments.

1- Introduction: please consider other literature that is missing about 3D using optical fiber sensores not only in silica fiber but also in polymer. Materials 11 (11), 2305, 2018; Sensors 19 (16), 3514, 2020. among others.

Answer: Introduction takes into account the possibility of using polymer optical fibers for testing. 4 new references have been added:

Leal-Junior, A.G.; Díaz, C.; Marques, C.; Frizera, A.; Pontes, M.J. 3D-Printing Techniques on the Development of Mul-tiparameter Sensors Using One FBG. Sensors 2019, 19, 3514, doi:10.3390/s19163514 (przenieść na 20 miejsce)

Leal-Junior, A.; Theodosiou, A.; Díaz, C.; Marques, C.; Pontes, M.J.; Kalli, K.; Frizera-Neto, A. Fiber Bragg Gratings in CYTOP Fibers Embedded in a 3D-Printed Flexible Support for Assessment of Human–Robot Interaction Forces. Materials 2018, 11, 2305. https://doi.org/10.3390/ma11112305 (przenieść na miejsce 21)

Leal-Junior, A.; Díaz, C.R.; Pontes, M.J.; Marques, C.; Frizera, A. Polymer optical fiber-embedded, 3D-printed instrumented support for microclimate and human-robot interaction forces assessment. Optics & Laser Technology, 2019, 112, 323-331, https://doi.org/10.1016/j.optlastec.2018.11.044

Zhao, C.; Xia, Z.; Wang, X.; Nie, J.; Huang, P.; Zhao, S. 3D-printed highly stable flexible strain sensor based on silver-coated-glass fiber-filled conductive silicon rubber. Materials & Design, 2020, 193, 108788, https://doi.org/10.1016/j.matdes.2020.108788

Fig. 8 to 10: please consider to add more cycles, increasing and decreasing cycles to confirm the repeatability. Also, Fig. 8a and Fig. 9a the FBG are the same? Why the spectra are so different? 

Answer:  The manuscript was supplemented with additional studies to confirm the repeatability (Fig. 11b). Figures 8a and 9a refer to two different samples, therefore the spectra are different.

 Fig. 11: it is not a good graph to expain the repeatability. It is just confirm the that spectrum comes to the initial position. Please comment.

Answer: The manuscript shows that the initial Bragg wavelength does not change after taking the measurements. Additional measurements added to the manuscript show that the measured strain values are constant over time.

Table 3. What is "nº 15"?  Also the principle for such different is not so clear.

Answer:  Tab. 3 has been edited.

How many probes were tested to guarantee the reproducibility between identical probes?

Answer:  The idea of the paper is to present an interesting solution only as a proof of concept, but not perform extended research necessary to create technical specifications. Hence there is a lack of full testing studies (not relevant for this stage yet).

Round 2

Reviewer 1 Report

From the authors' response, they claimed that there was no deformation induced to the optical fiber after the printing. However, they still did not explain the necessity of the 3D printing process. Would a bare optical fiber provide the same results? This reviewer does not recommend the publication of this paper in current form.

Author Response

Thank you very much for valuable and insightful comments. Fiber optic sensors belong to the most effective sensors suitable for material deformations monitoring. The main reason for this is that a fiber optic sensor becomes an integral part of the measured element and consequently there is no need to stick it outside, that would result in possible defects in such a connection. Unfortunately, since the optical fiber is very fragile and prone to damages, so in various solutions we must use buffering layers. In this case, 3D printing was used as a proposal for such a buffer. Our results obtained clearly indicate that such a solution is justified for certain applications. However, taking into account your comments, we have modified the description in the Introduction section:

Previous text:

Unfortunately, the main problem in published cases was the lack of protection of optical fiber, which could easily be damaged. By printing the fiber optic sensor into the structure of the compliant mechanism in order to prevent it from being damaged, the stiffness between the stretched arms will significantly increase. Usually, in compliant mechanisms the amplification of deformation amplitude takes place at the expense of tensile force. Too strong stiffening of the deformed arms precludes the use of elastic joints. Therefore, in this work we focused on designing a single compliant mechanism, which will allow for complete printing of the fiber optic sensor.

The text that takes into account the comments of the reviewer:

Unfortunately, the main problem in published cases was the lack of protection of optical fiber, which could easily be damaged [15,19]. Additionally, a significant weakness of such a system is a presence of adhesive connections due to material differences on the border between glue and PLA. One of the possible solutions to overcome this limitation in composite materials is to place the optical fiber inside the material structure. The main reason for this is that the sensor becomes an integral part of the measured element (smart structure) and consequently there is no need to stick it outside that would result in possible defects in this connection. Such a solution enables also protecting the most sensitive parts of the sensor against any mechanical damages and at the same time allows deformation measurement. Therefore, in this work we focused only on designing a single compliant mechanism, which allows for complete printing of the fiber optic sensor.

Reviewer 2 Report

Everything is ok. All necessary aspects were corrected.

Author Response

Thank you very much for valuable and insightful comments. 

Reviewer 3 Report

All my comments have been correctly addressed.

Author Response

(The authors gave the same response as above.)

Reviewer 4 Report

The paper is ready for publication

Author Response

(The authors gave the same response as above.)
